# Parental and Familial Factors Influencing Physical Activity Levels in Early Adolescence: A Prospective Study

**DOI:** 10.3390/healthcare8040532

**Published:** 2020-12-02

**Authors:** Dora Maric, Ivan Kvesic, Ivana Kujundzic Lujan, Antonino Bianco, Natasa Zenic, Vlatko Separovic, Admir Terzic, Sime Versic, Damir Sekulic

**Affiliations:** 1PhD Program in Health Promotion and Cognitive Sciences, Department of Psychology Educational Science and Human Movement, Sport and Exercise Sciences Research Unit, University of Palermo, 90100 Palermo, Italy; dora.maric@unipa.it (D.M.); antonino.bianco@community.unipa.it (A.B.); 2Faculty of Science and Education, University of Mostar, 88000 Mostar, Bosnia and Herzegovina; ivan.kvesic@fpmoz.sum.ba (I.K.); ivana.kujundzic.lujan@fpmoz.sum.ba (I.K.L.); 3Faculty of Kinesiology, University of Split, 21000 Split, Croatia; natasazenic@gmail.com (N.Z.); sime.versic@gmail.com (S.V.); 4Faculty of Sport Faculty of Physical Education and Sports, University of Tuzla, 75000 Tuzla, Bosnia and Herzegovina; vlatko.separovic@untz.ba (V.S.); admir.terza@bih.net.ba (A.T.)

**Keywords:** physical activity, parental factor, adolescents, gender differences

## Abstract

Parental/familial factors are important determinants of the physical activity level (PAL) in children and adolescents, but studies rarely prospectively evaluate their relationships. This study aimed to evaluate the changes in physical activity levels among adolescents from Bosnia and Herzegovina over a two-year period and to determine parental/familial predictors of PAL in early adolescence. A total of 651 participants (50.3% females) were tested at baseline (beginning of high school education; 14 years old on average) and at follow-up (approximately 20 months later). The predictors included sociodemographic characteristics (age, gender) and parental/familial factors (socioeconomic status of the family, maternal and paternal education, conflict with parents, parental absence from home, parental questioning, and parental monitoring). Physical activity levels were evidenced by the Physical Activity Questionnaire for Adolescents (PAQ-A; criterion). Boys were more active than girls, both at baseline (*t*-test = 3.09, *p* < 0.001) and at follow-up (*t*-test = 3.4, *p* < 0.001). Physical activity level decreased over the observed two-year period (*t*-test = 16.89, *p* < 0.001), especially in boys, which is probably a consequence of drop-out from the sport in this period. Logistic regression evidenced parental education as a positive predictor of physical activity level at baseline (OR [95% CI]; 1.38 [1.15–170], 1.35 [1.10–1.65]), and at follow-up (1.35 [1.11–1.69], 1.29 [1.09–1.59], for maternal and paternal education, respectively). Parents with a higher level of education are probably more informed about the importance of physical activity on health status, and thus transfer this information to their children as well. The age from 14 to 16 years is likely a critical period for maintaining physical activity levels in boys, while further studies of a younger age are necessary to evaluate the dynamics of changes in physical activity levels for girls. For maintaining physical activity levels in adolescence, special attention should be paid to children whose parents are less educated, and to inform them of the benefits of an appropriate physical activity level and its necessity for maintaining proper health and growth.

## 1. Introduction

Physical activity (PA) represents an important segment of physical and mental health status [1,2,3]. Irrespective of the importance of PA in all periods of life, reaching the appropriate PA level (PAL) is particularly important in childhood and adolescence, since regular PA is essential for healthy growth and development during this life period for a number of reasons. Young people with high PALs are less prone to cardiovascular diseases and type 2 diabetes, control their weight better, maintain a healthy musculoskeletal system and respiratory capacities, and experience mental health benefits (i.e., self-confidence and reduced likelihood of depression) and also have a higher level of self-perceived health status [4,5,6,7,8,9,10,11]. Moreover, a decreased PAL in adolescence can cause chronic health problems (e.g., cancer and cardiovascular and respiratory diseases) and obesity at a later age [12,13].

Although debates over the amount and type of PA needed for health benefits continue, the World Health Organization (WHO) recommends at least 60 min a day of moderate-to-vigorous PA for children and adolescents aged 5–17 years [14]. However, most children and adolescents worldwide do not meet the recommended PAL requirements [15,16,17,18]. Specifically, a global study found that only 20% of children between the ages of 13 and 15 exercise within the WHO recommendation levels [15]. These results differ among the studied populations; for example, only 15.9% of Americans between 14 and 17 years, 7% of Canadians between 6 and 19 years, and 5.6% of Chinese between 9 and 17 years engage in at least 60 min of moderate activity six times a week [16,17,18].

Adolescence is a critical period in an individual’s development, in which young people form attitudes about many important life issues, including health and educational habits, and the habits and attitudes that young people gain during this period will likely continue into the adult phase of their lives [1,19]. Despite the clearly emphasized and well-known importance of achieving appropriate PALs in young people, studies have shown that PALs significantly decline in adolescence [20].

In cross-sectional studies on youths in North America and Europe, research has indicated a clear decline of PA in adolescence [21,22]. Longitudinal studies have also noted a trend of decreased PALs in adolescents [4]. In a five-year longitudinal study, the authors tracked changes in habits from early to late adolescence and noticed a significant decline in weekly hours spent by young people engaged in PA; in contrast, they observed an increase in computer time, especially in boys [23]. Research on young Norwegians, conducted over a period of eight years (13–21 years of age), reported a decrease in PA between 13 and 19 years [24]. A recent study on older adolescents from Bosnia and Herzegovina detected a significant decrease in PALs between 16 and 18 years of age, irrespective of gender [25]. Obviously, there is a global trend of decreasing PALs; therefore, factors associated with such negative changes need to be elucidated.

Indeed, in order to solve this burning issue, a wide range of interventions have been developed with the aim of promoting the greater involvement of children and adolescents in some form of PA [9,20,26]. One of the most important preconditions for creating successful and targeted interventions is the identification of factors that affect PAL [27]. As a result, in recent years, a large number of studies have dealt with the analysis of the predictors of PALs in adolescents [20,28,29,30,31]. In general, PAL predictors can be divided into five groups: Demographic (biological) factors, psychological (cognitive) factors, behavioral characteristics, sociocultural factors, and the physical environment [32].

Globally, there are relatively consistent findings: (i) Boys are more active than girls, and (ii) PALs decrease during adolescence [33]. Furthermore, decreased PALs have been noticed in children from families with poorer socioeconomic status, simply because of the limited choice of physical activities/sports [34]. When it comes to psychological factors, studies have confirmed self-efficacy and motivation (which is primarily caused by perceived (sporting) competence and goal orientation) as the most important psychological predictors that positively affect PALs in young people [33]. Of the sociological factors, parental support stands out most consistently as a factor that positively influences PALs due to its positive influence on the involvement of youths in PA [28,35,36]. In this context, parents prove to be more important than other agents of socialization (colleagues, school, etc.) because, in addition to influencing young people as role models, they also serve as a kind of “gate keeper” by enrolling children in sports clubs, driving them to training, and so on [20].

Studies examining parental influence on children’s PAL have identified a number of predictors, including parental support, family connectedness, parental expectations, and parental monitoring [29,30,37,38,39]. Children of active mothers and fathers are proven to be multiple times more active than children of inactive parents [30]. Parental support was consistently positively and significantly associated with child PA in numerous studies examining these factors [29,39]. Among the overweight children and adolescents, higher levels of family connectedness, parental expectations and moderate levels of parental monitoring were associated with the lower level of PA [37]. Additionally, familial environmental and genetic factors showed to have a significant influence on the familial resemblance in PAL [38].

Despite all efforts, there is still no clear and definitive evaluation of the determinants of the PALs in adolescents [28]. On the one hand, this can be explained by the various measuring instruments used to estimate PA [28]. In brief, some authors used objective measurement techniques such as heart rate monitors, pedometers and accelerometers [40,41,42,43]. Meanwhile, most of the studies which analyzed PALs in children and adolescents used subjective technique methods including self-report questionnaires, interviewer-administered questionnaires, proxy-report questionnaires and diaries [44,45,46,47,48,49,50]. Self-report and interviewer-administered questionnaires rely on children’s self-reported activity in the past period, which can vary from the past three days to the whole year [44,45,46,47]. In proxy reports, parents and teachers provide information regarding children’s PAL [48,49]. Finally, diaries were used in only a few studies because it is very demanding for children to take regular notes about their PAL [50].

The results of previous studies were undoubtedly determined by oftentimes homogenous samples of participants, so the same factors need to be investigated in subgroups of different ethnicities, socioeconomic statuses, and environmental characteristics [28]. Moreover, the researchers that have studied this topic have highlighted regularly in their conclusions the need for a systematic and longitudinal analysis of a number of factors that, due to their complexity, should be observed together, analyzing cause–effect relationships between them, in order to obtain a more realistic picture of the determinants of PA during adolescence [36,51]. Although studies performed so far provided evidence on a decrease in PAL during the period of adolescence on the territory of southeastern Europe, there is a limited body of prospective evidence about: (i) changes in PAL which occur in younger adolescents, and (ii) factors which may influence such changes in this territory. Specifically, to the best of our knowledge no study so far prospectively examined the changes in PAL, while examining the socio-economic, socio-educational, and factors of parent–child relationship as covariates of changes in PAL in younger adolescents from the territory of former Yugoslavia.

For these reasons, the main aim of this study was to prospectively evaluate the changes in PALs among adolescents from Bosnia and Herzegovina over the 2-years period, between 14 and 16 years of age. Further, we evaluated the influence of socio-economic, socio-educational, and parent–child relationship factors on PALs at the beginning of the 1st year of high school (approximately 14 years of age), and at the end of the 2nd year of high-school (approximately 16 years of age). The authors hypothesized that a decline of PAL will occur during the course of the study. Additionally, we hypothesized that studied factors would influence PALs in both boys and girls.

## 2. Materials and Methods

### 2.1. Design and Participants

The participants in this prospective cohort study were adolescents from Bosnia and Hercegovina, more precisely, from Tuzla county, Herzegovina–Neretva county, and Western Herzegovina county. The sampling was based on a multi-stage cluster sampling method including (i) clustering of all schools from selected counties into two cluster (based on school-size), (ii) random sampling of 50% of high schools from each cluster, and (iii) random sampling of 50% of 1st grades from each of the selected schools. During the baseline testing, a total of 701 participants were examined. Therefore, the inclusion criteria for this study were: (i) regular participation in the high-school education in selected high-schools, and (ii) participation in testing at both testing waves (please see later for details on testing). No specific exclusion criteria were specified. At baseline (September 2017), the participants were at the beginning of the first grade of high school and were 14.3 ± 1.01 years old on average. The follow-up testing occurred 20 months later, in spring 2019, including 691 participants at the end of the second grade of high school. In this study, we included only those participants who were tested at both baseline and follow-up (*n* = 651; 50.3% females). Finally, the dropout rate was 12%. Included participants met the sample size criteria, since the required sample size for the observed population, and a level of significance of *p* < 0.05 was 398. The sampling procedure, drop-out rates and locations of the study are presented in Figure 1. 

### 2.2. Instruments

The variables in this study included: (i) Participants’ sociodemographic characteristics, (ii) parental/familial factors, and (iii) PALs. Sociodemographic characteristics and parental/familial factors were evaluated by structured questionnaires which were previously confirmed to be valid and reliable in similar samples of participants, and results are presented in detail elsewhere [52,53,54]. The sociodemographic variables included age (in years), and gender (male and female). The parental/familial factors observed in the study consisted of socioeconomic status of the family (responded on a three-point scale: below average, average, and above average), paternal and maternal education level (elementary school, high school, and college/university degree), conflict with parents (almost never, rarely, periodically, and often), parental absence from home (always at home, rarely absent, occasionally absent, and often absent), self-perceived parental care (parents do not care at all, do not care enough, good care, and very much care), and parental questioning about friends, school grades, problems, etc. (mostly never, rarely, from time to time, and often).

PALs were estimated with the Physical Activity Questionnaire for Adolescents (PAQ-A), which was previously validated and used in numerous studies [24]. The PAQ-A is a questionnaire form for which participants provide their self-reported PALs during the last seven days. The first eight questions contribute to the final score on a scale from 0 (minimum) to 5 (maximum PAL) refer to different forms of PA (e.g., sports, free play, and physical education in school). The last question was used only for the detection of injuries and/or illness that could have possibly prevented participants from PA in the last seven days. Finally, the difference between the PAQ-A results at baseline and those at the follow-up for each participant was calculated as a measure of changes in PALs. For the purpose of this study participants were grouped according to their PAQ-A results into two groups; those with sufficient/appropriate PAL (PAQ-A score of 2.73 and above), and those with insufficient/inappropriate PAL (PAQ-A score < 2.73) [26].

### 2.3. Procedures

All high schools in the selected counties were divided into two groups according to the number of children, and one-half of the first age classes were randomly selected from each of the groups. During the first school week (early September 2017), the examiners visited the schools and informed the children about the testing, as well as shared consent forms to participate in the research. The testing itself was conducted two weeks after, including only those students who brought a signed form by their parents. The study purposes and aims were explained to the students and their parents (in written form), and it was clearly indicated to them that the survey was strictly anonymous and that they could refuse to participate and/or not respond to any of the questions or to the whole questioning. Due to the specificity of testing at two-time points and for the purpose of pairing the results, children were asked to choose a personal anonymous code to use for both testing waves (i.e., the last three digits of their e-mail password). The survey lasted approximately 15 min, and was performed on an online internet platform, using the school equipment and resources or private mobile phones. Second testing was performed over the last two weeks of the school year 2018/19 (late May, early June 2019) using the same protocol. The entire testing procedure was performed in accordance with ethical guidelines and was approved by the Ethical Committee of the University of Split, Faculty of Kinesiology (approval number: 2181-205-05-02-05-14-005).

After conducting both baseline- and follow-up tests, analysis of the attrition bias was calculated No significant differences were evidenced in PAL between the children tested at both waves and the ones who dropped out. However, the drop-out rate was higher in males than in females. By calculating intracluster correlation (with schools as clusters), we evidence the relatedness of responses within each cluster (school) [55]. Specifically, the intracluster coefficient (IC) for the baseline PAL showed appropriate within-school variance (IC = 0.06).

### 2.4. Data Analysis

Descriptive statistics included calculation of means and standard deviations for PAQ-A, and frequencies and percentages for remaining variables.

The second phase of statistical processing included calculation of the differences between groups (boys vs. girls; sufficient PAL vs. insufficient PAL), and within groups (PAL at baseline vs. PAL at follow-up). Namely, *t*-test for dependent samples was used to evaluate the differences for PAL obtained at baseline and follow-up for the total sample, and stratified for gender. Differences between groups in raw PAL scores were evidenced by *t*-test for independent samples. Additionally, dichotomized PAL-values (insufficient-/sufficient-PAL) were compared between groups and this was completed by Chi-square (χ2) calculation. Mann–Whitney test (MW) was used to compare ordinal variables between groups, while χ2 was used to evidence the differences between groups in categorical variables. 

To define the influence of studied predictors on PAL at baseline and follow-up the logistic regression was applied. First, each predictor was correlated with dichotomized PAQ-A values (insufficient PAL was coded as “1”, and sufficient PAL was coded as “2”). In order to further evaluate the eventual co-variability of the predictors, and to identify/eliminate any possible causal relationship between predictors, in the last phase of the statistical analyses the multivariate logistic regressions were calculated for dichotomized criteria (PAL at baseline, and PAL at follow-up). For such purpose, all predictors found to be significantly correlated to PAL were included in the multivariate logistic regression model. The final model was checked by the Hosmer–Lemeshow test of model fit (with significant χ2 indicating inappropriate model fit). Negelkerke R square, *p*-values, Odds Ratio (OR) and corresponding 95% Confidence Interval (95% CI) were reported as indicators of association between predictors and criteria.

## 3. Results

PALs decreased significantly during the course of the study in total sample (from 2.26 ± 1.13 to 2.13 ± 1.06; *t*-test = 16.89, *p* < 0.001), and when observed separately for boys (from 2.42 ± 1.19 to 2.28 ± 1.01, *t*-test = 10.41, *p* < 0.001), and for girls (from 2.14 ± 1.07 to 2.01 ± 0.99, *t*-test = 13.42, *p* < 0.001). The PAL was higher in boys than in girls at baseline (*t*-test = 3.09, *p* < 0.001), and at follow-up (*t*-test = 3.4, *p* < 0.001).

The 31% of adolescents reached appropriate/sufficient PAL at baseline (38% boys and 26% girls), while only 26% of them had appropriate PAL at follow-up (31% girls and 22% girls) (Figure 2). When observed at categorical scale (sufficient/insufficient PAL) the differences between genders were significant at baseline (χ2 = 9.54, *p* < 0.001), and at follow-up measurement (χ2 = 7.08, *p* < 0.01), with a higher prevalence of sufficient PAL among boys.

The differences between adolescents who achieved and those who did not achieve sufficient PAL are presented in Table 1. Parental monitoring was lower in adolescents with insufficient PAL at baseline (MW Mann Whintey test = 2.12, *p* = 0.03). A sufficient PAL at baseline was found in children whose fathers and mothers were better educated (MW = 2.74, *p* < 0.01 and MW = 3.3, *p* < 0.001 for paternal and maternal education, respectively).

The higher maternal- (MW = 2.72, *p* < 0.01), and paternal-education (MW = 2.13, *p* < 0.05) was found in adolescents who had appropriate PAL at follow-up. No significant differences between groups based on PAL were evidenced for other predictors (Table 2).

Figure 3 and Figure 4 present univariate relationships between baseline sociodemographic and parental/familial factor, and dichotomized PAL criteria at baseline (Figure 3) and at follow-up (Figure 4). At baseline, the higher likelihood for appropriate PAL was found in males (Negelkerke R square: 0.02; OR: 1.68, 95% CI: 1.21–2.34; *p* < 0.001), for those adolescents whose mothers (Negelkerke R square: 0.02; OR: 1.38, 95% CI: 1.15–1.70; *p* < 0.001), and whose fathers were better educated (Negelkerke R square: 0.02; baseline: OR: 1.35, 95% CI: 1.10–1.65, *p* < 0.01). At follow-up higher likelihood for appropriate PAL was found for boys (Negelkerke R square: 0.02; OR: 1.54, 95% CI: 1.11–2.03, *p* < 0.001), adolescents who reported better maternal (Negelkerke R square: 0.02; OR: 1.35, 95% CI: 1.11–1.69, *p* < 0.05), and those who reported better paternal-education (Negelkerke R square: 0.015; OR: 1.29, 95% CI: 1.09–1.59, *p* < 0.05).

Multivariate logistic were calculated while simultaneously including all variables evidenced as being significant univariate predictors of PAL at baseline and follow-up. Male gender (OR: 1.55, 95% CI: 1.11–1.91, *p* < 0.001), higher paternal education (OR: 1.35, 95% CI: 1.05–1.67, *p* < 0.05), and higher maternal education (OR: 1.34, 95% CI: 1.06–1.71, *p* < 0.05) were all significantly related to PAL-baseline (Negelkerke R square: 0.04). In total 67% of the participants were correctly classified according to specified regression function. A similar multivariate relationship was found when PAL at follow-up was observed as a criterion. Namely, higher paternal education (OR: 1.21, 95% CI: 1.01–1.44, *p* < 0.05), and higher maternal education (OR: 1.30, 95% CI: 1.05–1.57, *p* < 0.01), together with male gender (OR: 1.50, 95% CI: 1.05–2.15, *p* < 0.01) were positively correlated with appropriate/sufficient PAL at follow-up (Negelkerke R square: 0.05), with 71% participants being correctly classified. Results of the multivariate logistic regressions actually evidence the independent influence of paternal education and gender on PAL (Figure 5). A Hosmer–Lemeshov test evidenced appropriate model fit for multivariate logistic regression models calculated for PAL at baseline (χ2 = 7.37 *p* = 0.39), and for PAL at follow-up (χ2 = 8.01, *p* = 0.31).

## 4. Discussion

There are several important findings of this study. First, boys were more active than girls, PALs decreased over the study course, and the decrease was more evident in boys. As a result, we may support our initial study hypothesis. Second, parental education was evidenced as a positive influencing factor of PALs in both testing waves. Finally, no significant influences of socioeconomic status, parental/familial conflict, parental absence from home, parental care, and parental questioning on PALs were found. Therefore, our second hypothesis may be partially accepted.

### 4.1. Changes and Differences in PALs

Our results evidenced higher PALs in boys than in girls in early adolescence (between 14 and 16 years of age). Such findings are in line with previous research, which, almost without exception, reported higher PALs in boys [4,21,22,28]. Specifically, an epidemiological review article that analyzed PALs and fitness levels in children and adolescents concluded that boys are approximately 14% and 25% more active than their female peers, respectively, and that over the school years, PALs decrease by 3–7% [56]. Previous studies have highlighted three of the most common perspectives for explaining the important determinants of gender-differences in PALs: (i) Socialization, (ii) attitudinal factors, and (iii) organized sports [30,57,58,59,60].

In the context of socialization, research has highlighted the significant impact of the environment (i.e., parents, peers, and teachers) on PALs in adolescents [30,58]. This impact differs between genders, which is explained by an interaction between gender and parental or teacher involvement [57]. Additionally, a larger proportion of girls have negative experiences with practicing some form of physical activity and sports (e.g., feeling stupid or incompetent, being negatively evaluated, not having enough choice, and using inadequate facilities), which consequently reduces the level of their involvement in physical activity and sports [60].

An attitudinal explanation of the gender differences in physical activity and sports involvement assumes that gender roles foster differences in attitudes that contribute to differences in practicing sports and in physical activity in general [57]. Among other things, a greater association has been noticed between masculine identities and sports. In other words, boys are more competitive and generally more interested in sports than girls, which consequently contributes to their higher PALs [61,62].

Irrespective of previous explanations, research has highlighted organized sports as a key factor in explaining gender differences in PALs [57]. This perspective is based on the fact that a large proportion of the PALs among young people refers to organized sports in sports clubs, where have much greater gender differences in inclusion compared to the overall gender difference in physical activity [59]. In short, the organized sports system favors men through organizational specifics, more competition, and more accessible sports facilities, and also the dropout rate for girls is much higher, especially taking into account girls’ menstrual cycle and more frequent absence from sports training in girls due to menstrual pain and/or hygienic reasons [63,64]. Put together, the higher PALs among boys is understandable.

The decrease in PALs in our sample is in accordance with previous studies, where authors have regularly confirmed a decrease in PALs during adolescence [20,56]. More specifically, a decline in PALs from the beginning of the first grade of high school to the end of the second grade was approximately 7%, and this was more emphasized in boys (8%) than in girls (5%). Several studies have analyzed longitudinal PAL changes during adolescence, and have evidenced various causes for the decrease in PALs [23,65]. Collectively, authors most often point out the increase in school obligations, weight gain, and increased screen time, as well as a decline in active transport.

Although not entirely consistent, previous research has more often identified a higher rate of PAL decline in boys during younger adolescence [66,67]. However, it must be observed in light of the higher PALs in boys than in girls at the beginning of the observed period [57,68]. The reason for this stands in the fact that girls drop out of sports earlier (between 9 and 12 years). Therefore, during younger adolescence, girls generally have a lower baseline level of PA and thus less of a chance of an additional decline in PALs than boys, who, during this period (13–16 years), start to drop out from organized sports [67].

### 4.2. Parental Education and Physical Activity during Adolescence

Thus far, much research has analyzed the influence of parental and familial factors on PALs in children and adolescents [35,36,69]. Above all, family support has been consistently shown to be a positive predictor of PALs in adolescents [35]. Specifically, parents have been proven to be one of the most important predictors of PALs, because they serve as role models and they finance and actively participate in the organization of sports activities for their children [20]. However, previous studies have not consistently highlighted parental education as a predictor of PALs in adolescence.

Some studies have shown that there is no significant influence [35,69], while others have indicated a positive influence of a higher educational level of parents on their children’s PALs [70,71]. In studies that have confirmed a positive influence, this fact was primarily explained by the relationship between the level of education and familial socioeconomic status and income [70]. For example, a longitudinal study on 1213 African-American and 1166 Caucasian girls found a significant impact of parental education on PALs, which was explained by the negative impact of low socioeconomic status and a potentially stressful home environment on PALs in children, highlighting the need to solve social disparities that potentiate health disparities [70].

In contrast, although parental education was positively correlated with PALs in children, we found no significant effect of socioeconomic status on PALs, and therefore our results are not absolutely consistent with previous findings [36,72]. Before discussing it must be noted that data on socio-economic status for adolescents observed herein could be observed as plausible since results are comparable to data reported in previous studies where somewhat older adolescents from the same country were included [53,73]. Most likely, in the region where the study was conducted, the financial status of the family does not have a strong influence on sports involvement as in some other regions of the world. Specifically, in studied counties, as well as on the whole territory of Bosnia and Herzegovina, the majority of sports are available to all children (i.e., participation is mostly free), the distances between home and sports facilities are relatively short (i.e., children do not depend on parental transport), and most popular sports do not require specific and expensive sports equipment, and are practiced in school facilities and gyms (i.e., team sports like football, handball, basketball) [52]. Taken together, even the influence of socioeconomic status on participation in sports and PA in the studied adolescents was reduced. Therefore, the cause of the connection between parental education and the children’s PALs in our study should be found in something else.

Mainly, it is reasonable to assume that parents with a higher level of education are generally better informed and are more aware of the importance of PA on the health status of their children [27]. Therefore, better-educated parents more likely to encourage their children to engage in some form of PA or sports. This explanation is in accordance with a recent discussion offered in a study where maternal education was positively correlated with PALs in somewhat older adolescents, and where the authors highlighted mothers as being more involved in their children’s life in later adolescence, even in the domain of physical activity, irrespective of the known influence of fathers on sports participation, which is more evident in earlier adolescence [27]. Interestingly, even in our study the effect size (based on R square from univariate logistic regressions) for PAL at follow-up was higher for maternal education, than for paternal education (Nagelkerke R square = 0.02 and 0.015, for maternal- and paternal-education, respectively) which indicate the stronger influence of maternal education on PAL of the children at the age of 16. Meanwhile, based on the same statistical parameter, maternal- and paternal-education are equally important predictors of PAL at baseline (i.e., 14 years of age).

### 4.3. Familial Variables and Physical Activity in Adolescence

Parental and familial factors are known to be important determinants of various health-related behaviors in adolescents, and previous studies have regularly confirmed the direct influence of family cohesion, parent–child relationships, and parental involvement in children’s daily activities on PALs [74,75]. These dimensions of parental influence may be a reflection of an authoritative parenting style, which generally includes reaction, in the form of providing emotional support and involvement, but also demanding in terms of providing an appropriate level of parental control [75]. Studies find that authoritative parenting styles are often associated with higher levels of student achievement [76]. Furthermore, a study examining PALs and sedentary lifestyles in girls identified an authoritative parenting style as a significant positive predictor of PALs [77].

Similar findings have been found in studies of some other issues, such as smoking [78]. Specifically, it has been shown that parents with an authoritative parenting style who have anti-smoking household rules and who do not smoke themselves are more likely to have adolescents who do not consume cigarettes [78]. Additionally, in the study examining optimal parent–child relationships, parental warmth and strictness were highly associated with a child’s well-being [79]. However, in our study, the variables examining parent–child relationships, familial conflict, parental absence from the home, parental care, and parental questioning were not significant predictors of PALs.

The most likely reason for such relative inconsistency in our results (i.e., non-significant influence of parental control and monitoring variables on the PALs of their children) compared to those reported previously, where the authors regularly confirmed significant correlations between similar sets of variables that could be found in the differences between the established “magnitude” of parental control/monitoring. Namely, in our study, a minority of the children reported “low levels of parental control/monitoring” (see Results for details). Meanwhile, when using an identical measurement tool and studying somewhat older adolescents, previous reports from the same region (i.e., southeastern Europe) have shown a considerably larger variance in their results [54]. It could simply mathematically result in a higher possibility of reaching a statistically significant association between variables [80].

### 4.4. Limitations and Strengths

The first limitation of the study is the subjective nature of the measurement tool, since data were collected by a questionnaire. Therefore, there is a possibility that the participants did not answer honestly and/or tended to provide socially desirable answers. However, the authors strongly believe this possibility was reduced due to the testing protocol, the experience of the researchers, and the strict anonymity of the testing. Moreover, PALs were evidenced by questionnaire and were not objectively measured. However, the use of objective measures of PALs (e.g., accelerometers, pedometers) was limited because of the large sample size and the prospective nature of the study. Next, the observed sample did not represent the whole country, so the results should be generalized only for specific regions. Finally, this study did not take into account biological maturity, but only chronological maturity of the adolescents. Considering the well-known influence of maturity on multiple factors, future investigations should pay attention to it, and include the biological age in analyses of such kind.

This is one of the first studies that systematically examined the parental/familial predictors of PALs in adolescents from southeastern Europe. Additionally, this is probably the first prospective study of PAL changes and the predictors of PALs in young adolescents in this region. One of the strengths of this study is the fact that the tested sample was twice as large as that theoretically required for high statistical power, with a minor dropout rate. Finally, given that decreased PALs are highlighted globally as a major public health concern because of their association with the leading causes of death, illness, and disability, the authors believe that the findings of this research can be used in the global fight against the pandemic of physical inactivity, and that they will initiate further research. Findings of this study suggest practical interventions on children with parents of a lower educational level and the need to further examine the decline in PAL for girls at an earlier age than analyzed here. Moreover, additional factors related to decreased PAL in boys between 14 and 16 years of age should be evaluated.

## 5. Conclusions

The results of this study showed that boys are generally more active than girls in the period of younger adolescence (14–16 years). With a prospective follow-up over a two-year period, a declining trend in PALs was observed, and this negative trend was larger in boys. Given that previous researchers have found an earlier dropout from organized sports in girls (between 9 and 12 years old), it is obvious that the period between 14 and 16 years of age is particularly important for boys. Future studies should certainly investigate all of the factors that lead to dropouts from sports in younger male adolescents and the predictors of PAL decline in girls should be studied in younger years.

This study analyzed parental factors as possible indicators of PALs in younger adolescents, and especially emphasized the education of parents in both testing waves as a positive predictor. Therefore, in order to prevent a decrease in PALs in early adolescence, there is an evident need to specifically focus on children whose parents are of a lower educational level. This will hopefully improve their awareness of the importance of PA in this period of life. Such efforts will consequently have an important positive impact on the overall health status both in adolescence and later life.

Additionally, the established influence of parental education on the PAL of their children should be observed out of the context of the relationship. Namely, we can anticipate that specific education of the parents about the benefits and importance of physical activity would be directly translated to PAL of their children. Therefore, we may encourage studies that will investigate the effects of education of the parents (i.e., responsible adults) on their children’s PAL. In doing so special attention should be placed on parents of lower educational status and health-related topics of physical activity.

Parental/familial variables explaining socioeconomic status, as well as parental/familial control and monitoring, were not shown to be important predictors of PALs in the studied period (between 14 and 16 years of age). Most likely, the fact that the studied adolescents evidenced low levels of parental conflict and high parental control reduced the possibility that these variables significantly influenced PALs. Moreover, in the studied region, sports activities are relatively available to all children, which reduced the possibility that the socioeconomic status of the studied adolescents significantly contributed to PALs.

## Figures and Tables

**Figure 1 healthcare-08-00532-f001:**
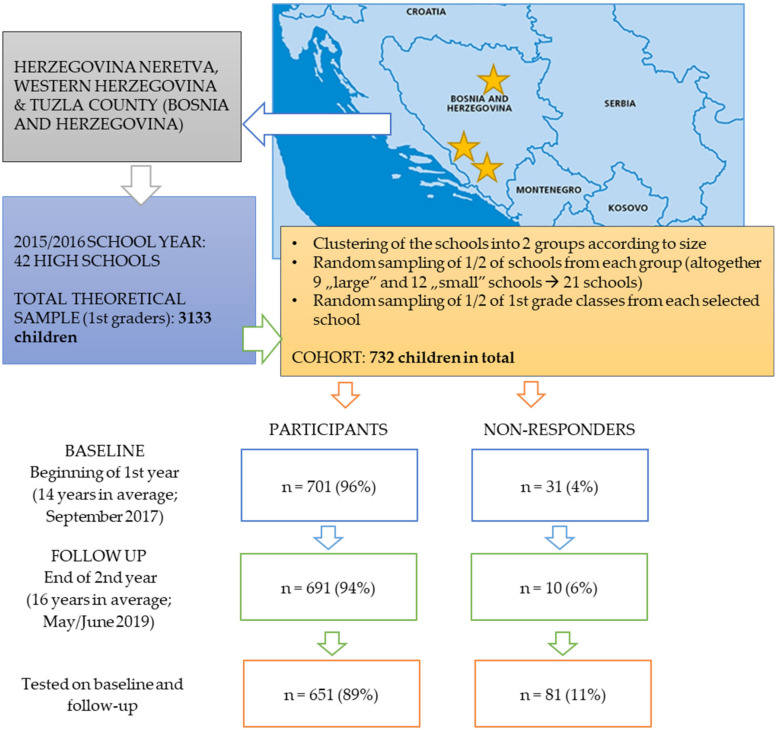
Study location, number of tested participants over testing waves, and drop-out rates.

**Figure 2 healthcare-08-00532-f002:**
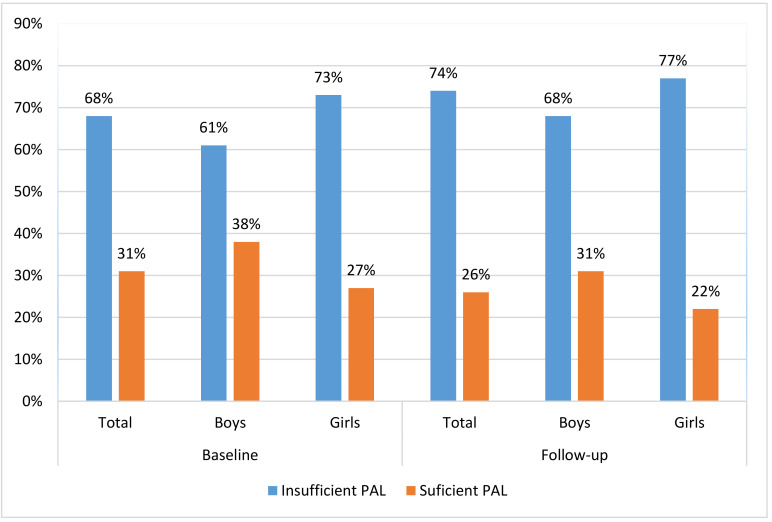
Prevalence of insufficient/sufficient physical activity level (PAL) in adolescents from Bosnia and Herzegovina at baseline (beginning of high school education) and follow-up (end of 2nd year of high-school).

**Figure 3 healthcare-08-00532-f003:**
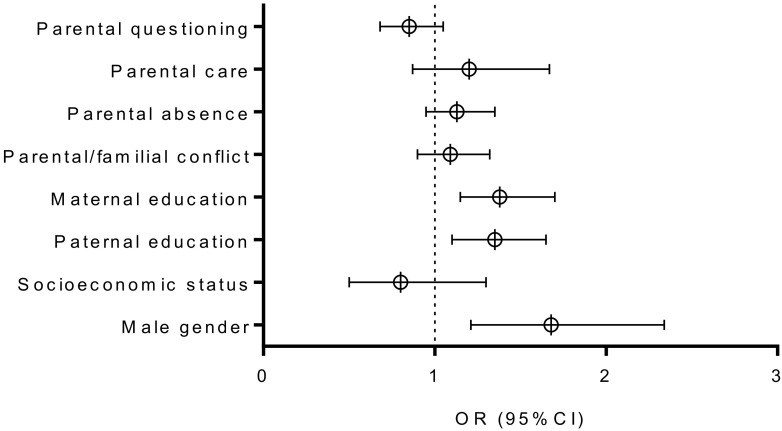
Correlates of sufficient physical activity level at baseline (results are presented as Odds Ratio [OR]± 95% Cofidence Interval [CI]; dotted line presents OR of 1 Odds Ratio and statistical significance of *p* < 0.005 if not crossed by 95% CI bar).

**Figure 4 healthcare-08-00532-f004:**
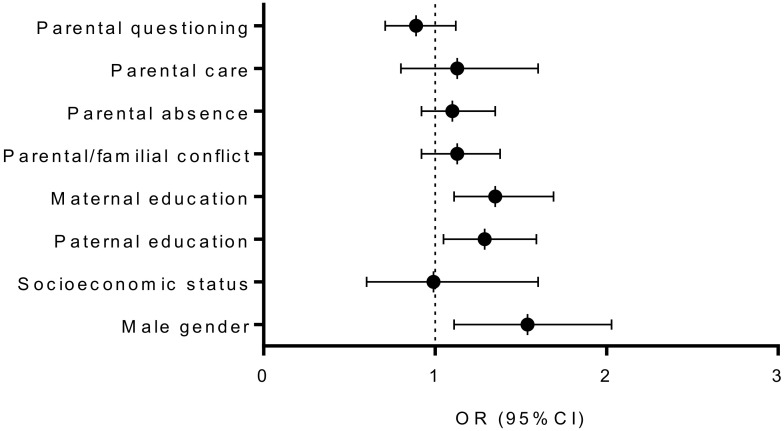
Correlates of sufficient physical activity level at follow-up (results are presented as Odds Ratio [OR]± 95% Cofidence Interval [CI]; dotted line presents OR of 1 Odds Ratio and statistical significance of *p* < 0.005 if not crossed by 95% CI bar).

**Figure 5 healthcare-08-00532-f005:**
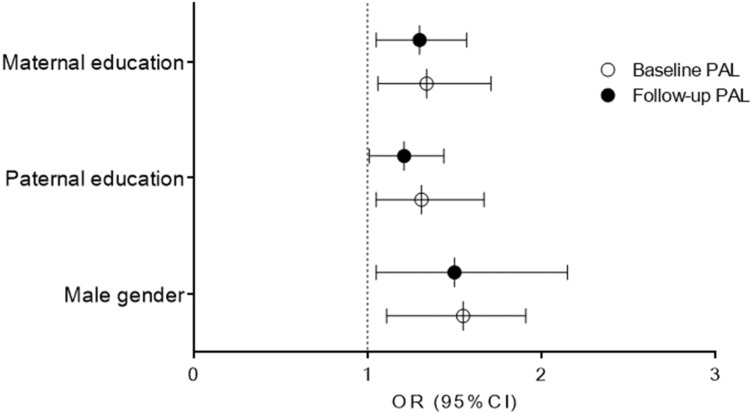
Multivariate logistic regression correlates of sufficient physical activity level (PAL) at baseline and at follow-up measurement (results are presented as Odds Ratio [OR]± 95% Cofidence Interval [CI]; dotted line presents OR of 1 Odds Ratio and statistical significance of *p* < 0.005 if not crossed by 95% CI bar).

**Table 1 healthcare-08-00532-t001:** Differences between adolescents grouped according to sufficiency/insufficiency of physical activity level (PAL) at baseline.

Variables	Baseline PAL				Mann–Whitney	
	Sufficient		Insufficient		MW	*p*
	F	%	F	%		
Socioeconomic status					1.09	0.27
Below average	2	0.9	8	1.8		
Average	188	87.9	372	81.8		
Above average	17	7.9	55	12.1		
Paternal education					2.74	0.01
Elementary school	2	0.9	11	2.4		
High school	110	51.4	264	58.0		
College level	48	22.4	96	21.1		
University level	47	22.0	63	13.8		
Maternal education					3.3	0.001
Elementary school	3	1.4	14	3.1		
High school	103	48.1	265	58.2		
College level	51	23.8	91	20.0		
University level	50	23.4	68	14.9		
Parental/familial conflict					1.06	0.28
No. never	157	34.5	66	30.8		
Rarely	188	41.3	91	42.5		
From time to time	72	15.8	42	19.6		
Often	24	5.3	10	4.7		
Parental absence					1.42	0.15
Always at home	40	18.7	103	22.6		
Rarely absent	72	33.6	156	34.3		
Occasionally absent	76	35.5	136	29.9		
Frequently absent	21	9.8	42	9.2		
Parental care					0.61	0.53
Do not care at all	0	0.0	3	0.7		
Do not care enough	3	1.4	15	3.3		
Good care	43	20.1	85	18.7		
Very much care	163	76.2	333	73.2		
Parental questioning					2.12	0.03
Mostly never	4	1.9	12	2.6		
Rarely	18	8.4	41	9.0		
From time to time	102	47.7	158	34.7		
Often	84	39.3	228	50.1		

Legend: Note that participants were not obligated to respond to all questions and therefore for some variables all responses summarized does not equal the total sample of participants (*n* = 651).

**Table 2 healthcare-08-00532-t002:** Differences between adolescents grouped according to sufficiency/insufficiency of physical activity level (PAL) at follow-up.

Variables	Follow-Up PAL				Mann–Whitney	
	Sufficient		Insufficient		MW	*p*
	F	%	F	%		
Socioeconomic status					0.24	0.8
Below average	1	0.6	9	1.8		
Average	154	86.5	406	82.7		
Above average	16	9.0	56	11.4		
Paternal education					2.13	0.03
Elementary school	1	0.6	12	2.4		
High school	94	52.8	280	57.0		
College level	37	20.8	107	21.8		
University level	39	21.9	71	14.5		
Maternal education					2.72	0.01
Elementary school	2	1.1	15	3.1		
High school	89	50.0	279	56.8		
College level	35	19.7	107	21.8		
University level	45	25.3	73	14.9		
Parental/familial conflict					1.47	0.14
No. never	51	28.7	172	35.0		
Rarely	78	43.8	201	40.9		
From time to time	36	20.2	78	15.9		
Often	8	4.5	26	5.3		
Parental absence					1.24	0.21
Always at home	31	17.4	112	22.8		
Rarely absent	62	34.8	166	33.8		
Occasionally absent	65	36.5	147	29.9		
Frequently absent	15	8.4	48	9.8		
Parental care					0.32	0.75
Do not care at all	0	0.0	3	0.6		
Do not care enough	3	1.7	15	3.1		
Good care	36	20.2	92	18.7		
Very much care	134	75.3	362	73.7		
Parental questioning					1.44	0.15
Mostly never	3	1.7	13	2.6		
Rarely	16	9.0	43	8.8		
From time to time	81	45.5	179	36.5		
Often	73	41.0	239	48.7		

Legend: Note that participants were not obligated to respond to all questions and therefore for some variables all responses summarized does not equal the total sample of participants (*n* = 651).

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
