# Peer review of "Parental and Familial Factors Influencing Physical Activity Levels in Early Adolescence: A Prospective Study"

_healthcare, 2020, doi:10.3390/healthcare8040532_

Round 1

Reviewer 1 Report

This is a very well written and easy to follow manuscript.  I only have a few concerns.

Major concern:

Tables 1 and 2.  Does "F" stand for sample size ("n")?  If so, for example, 642 participants responded to Socioeconomic status versus 651 total.  Table 1, % sufficient should be 1, 91 and 8, total 100% versus 96.7% as presented. Please clarify what "F" and "%" signify and why the % does not equal 100.  If I am correct in assuming that F = n, and % = n/total responses, then each section (e.g. socioeconomic status, paternal education, etc.) should be adjusted accordingly.   

Minor concerns:

Materials and Methods, line 112.  Change "were and" to "and were."

Figure 2.  A line between Insufficient and Sufficient PAL results would be helpful.

Tables 1 and 2.  How does the percent of socioeconomic status responders relate to the percent of socioeconomic status in the three counties?  It may be helpful to consider the similarities or discrepancies of the sample population to the total population in the Discussion.

Author Response

Comments and Suggestions for Authors

This is a very well written and easy to follow manuscript.  I only have a few concerns.

Thank You for your support and review. We followed your comments and suggestions and amended manuscript accordingly. Please find all answers bellow.

Major concern:

Tables 1 and 2.  Does "F" stand for sample size ("n")?  If so, for example, 642 participants responded to Socioeconomic status versus 651 total.  Table 1, % sufficient should be 1, 91 and 8, total 100% versus 96.7% as presented. Please clarify what "F" and "%" signify and why the % does not equal 100.  If I am correct in assuming that F = n, and % = n/total responses, then each section (e.g. socioeconomic status, paternal education, etc.) should be adjusted accordingly.  

Response: Thank You for your comment. The “F” stands for frequency and “%” for percentage of whole sample in each answer category. For some variables, certain respondents did not answer the question and therefore there is a disparity between the total number of answers and the total number of study participants. However, it is clear that we had to pay attention on it. It is now clearly stated in the Table legends (please see legends for Table 1 and 2; thank you)

Minor concerns:

Materials and Methods, line 112.  Change "were and" to "and were."

Response: Thank You for noticing. The text is amended accordingly.

Figure 2.  A line between Insufficient and Sufficient PAL results would be helpful.

Response: Amended accordingly; line between categories is added

Tables 1 and 2.  How does the percent of socioeconomic status responders relate to the percent of socioeconomic status in the three counties?  It may be helpful to consider the similarities or discrepancies of the sample population to the total population in the Discussion.

Response: Thank you for this suggestion. Indeed, this is important information and we included brief comparison with previous studies. In short, it seems that data of socioeconomic status of here studied children are comparable to other data from the country. It is briefly presented and text reads: “Before discussing it must be noted that data on socio-economic status for adolescents observed herein could be observed as plausible since results are comparable to data reported in previous studies where somewhat older adolescents from the same country were included [58,59].” (please see 3rd paragraph of the subsection 4.2; thank you).

Thank you for your suggestions and comments.

Staying at your disposal

Reviewer 2 Report

  In the first place I would like to share the need to carry out work like the one you present. They are necessary for the advancement of advanced society. It is necessary to know the existing relationships between Parental / family factors are important determinants to be able to know and determine the level of physical activity of those involved. I would like to show that the instrument used is highly supported by the scientific community and provides objective, contrasted data to be able to be replicated. I think the treatment of the discussion divided by sections has been a great success. It facilitates the follow-up of the reader before the questions posed throughout the manuscript. Perhaps the limitations of the study should be reflected more clearly, if there was any inclusion or exclusion of the sample and the reasons. And finally a question of interest, if at some point the authors have taken into account the maturation stage of preadolescents, since the chronological age of men and women in this phase is only a piece of information. It is known to the scientific community that the psychosocial and cognitive factors of women are different.  

Author Response

In the first place I would like to share the need to carry out work like the one you present. They are necessary for the advancement of advanced society. It is necessary to know the existing relationships between Parental / family factors are important determinants to be able to know and determine the level of physical activity of those involved. I would like to show that the instrument used is highly supported by the scientific community and provides objective, contrasted data to be able to be replicated. I think the treatment of the discussion divided by sections has been a great success. It facilitates the follow-up of the reader before the questions posed throughout the manuscript. Perhaps the limitations of the study should be reflected more clearly, if there was any inclusion or exclusion of the sample and the reasons. And finally a question of interest, if at some point the authors have taken into account the maturation stage of preadolescents, since the chronological age of men and women in this phase is only a piece of information. It is known to the scientific community that the psychosocial and cognitive factors of women are different. 

Response:

We must thank you for recognizing the importance and quality in our work. By all means we share your opinion about the importance of the topic, and will certainly try to investigate it in the future. With regard to your comment and suggestion we may say as follows.

Yes, problem of maturity is definitively important, but at the same time, the prospective study such as this one we have performed are limited with regard to such approach. Also, and as you can see, some questions may be observed as “intimate”, and measurement of the maturity will make the study additionally complicated. Therefore, in this investigation was didn’t taken into account the maturation stage of preadolescents, but considering scientifically confirmed maturity influence on multiple factors (i.e. psychosocial and cognitive) this is something that should definitely be included in future similar studies. This is clearly stated in the Limitations section and text reads: “Finally, this study did not take into account biological maturity, but only chronological maturity of the adolescents. Considering the well-known influence of maturity on multiple factors, future investigations should pay attention on it, and include the biological age in analyses of such kind.” (Please see end of 1st paragraph subsection 4.-4. Limitations and strengths)

Thank you for your suggestions and comments.

Staying at your disposal

Reviewer 3 Report

Lines 61-63: add a reference

Lines 74-77: add a reference

Lines 77-78: you mention ‘a large number of studies’ but cite only two studies

Lines 93-94: add a reference

Lines 94-95: add a reference

Line 106: what is your hypothesis?

Line 108: how were the subjects recruited? What were the criteria for inclusion and exclusion?

Line 230: I think that maternal education is a part of parental education?

Lines 251-253. You mention ‘studies’ but cite only one study

Line 265: I suggest replacing ‘logically’ by ‘consequently’

Lines 307-309: add a reference

Lines 309-313: add a reference

Line 334: add a reference

Lines 340-344: add a reference

Line 368: add the practical applications and the implications for future research

Author Response

Thank you for your suggestions. Please bellow for our amendments. 

Lines 61-63: add a reference

Response: Thank You for noticing. The reference was added:

- Sterdt, E.; Liersch, S.; Walter, U. Correlates of physical activity of children and adolescents: A systematic review of reviews. Health Education Journal 2014, 73, 72-89.

Lines 74-77: add a reference

Response: Thank you for your suggestion. In the revised version of the manuscript we included following references for specified text.

  • Salmon, J.; Booth, M.L.; Phongsavan, P.; Murphy, N.; Timperio, A. Promoting physical activity participation among children and adolescents. Epidemiologic reviews 2007, 29, 144-159
  • Baranowski, T.; Anderson, C.; Carmack, C. Mediating variable framework in physical activity interventions: How are we doing? How might we do better? American journal of preventive medicine 1998, 15, 266-297.
  • Sekulic, D.; Rodek, J.; Sattler, T. Factors associated with physical activity levels in late adolescence: a prospective study. Medycyna Pracy 2020, 71.

Specifically, text now reads: “Indeed, in order to solve this burning issue, a wide range of interventions have been developed with the aim of promoting the greater involvement of children and adolescents in some form of PA [9,20,26]. One of the most important preconditions for creating successful and targeted interventions is the identification of factors that affect PAL [27].” (please see 5th paragraph of the Introduction).

Lines 77-78: you mention ‘a large number of studies’ but cite only two studies

Response: Thank You for your comment. Cited studies are reviews that included multiple studies on physical activity topic. However, we added new references:

  • Trost, S.G.; Loprinzi, P.D. Parental influences on physical activity behavior in children and adolescents: a brief review. American Journal of Lifestyle Medicine 2011, 5, 171-181.
  • Moore, L.L.; Lombardi, D.A.; White, M.J.; Campbell, J.L.; Oliveria, S.A.; Ellison, R.C. Influence of parents' physical activity levels on activity levels of young children. The Journal of pediatrics 1991, 118, 215-219.
  • Lubans, D.R.; Foster, C.; Biddle, S.J. A review of mediators of behavior in interventions to promote physical activity among children and adolescents. Preventive medicine 2008, 47, 463-470.

Text now reads: “As a result, in recent years, a large number of studies have dealt with the analysis of the predictors of PALs in adolescents [20,28-31].” (Please see 5th paragraph of the Introduction)

Lines 93-94: add a reference

Response: The reference was added:

  • Sallis, J.F.; Prochaska, J.J.; Taylor, W.C. A review of correlates of physical activity of children and adolescents. Medicine and science in sports and exercise 2000, 32, 963-975.

Text reads: “However, a review of the literature shows that the results are quite inconsistent, and thus, there is no clear and definitive evaluation of the determinants of the PALs in adolescents [28].” (please see beginning of the 7th paragraph of the Introduction)

Lines 94-95: add a reference

Response: The reference was added and is the same as line above.

  • Sallis, J.F.; Prochaska, J.J.; Taylor, W.C. A review of correlates of physical activity of children and adolescents. Medicine and science in sports and exercise 2000, 32, 963-975.

Text now reads: “On the one hand, this can be explained by the various measuring instruments used to estimate PA [28].”

Line 106: what is your hypothesis?

Response: Thank You for your comment. The hypothesis is added and text now reads: “Authors hypothesized that decline of PAL will occur during the course of the study. Additionally, we hypothesized that parental/familial factor would influence PALs in both boys and girls.” (please see end of the Introduction). Also, we responded to our hypotheses at the very beginning of the Discussion section (please see first paragraph of the Discussion)

Line 108: how were the subjects recruited? What were the criteria for inclusion and exclusion?

Response: Thank you for your question. We must agree that initially we didn’t explain the recruitment in sufficient details. In this version of the manuscript it is more clearly stated and now reads. “The participants in this study were adolescents from Bosnia and Hercegovina, more precisely, from Tuzla county, Herzegovina–Neretva county, and Western Herzegovina county. The sampling was based on a multi-stage cluster sampling method including (i) clustering of all schools from selected counties into two cluster (based on school-size), (ii) random sampling of 50% of high schools from each cluster, and (iii) random sampling of 50% of 1st grades from each of the selected schools.” (Please see beginning of the 1st paragraph of the subsection 2.1. Design and participants; thank you!)

Line 230: I think that maternal education is a part of parental education?

Response: Thank You for noticing. It was misspelled “parental” instead of “parental”. The text is amended accordingly and now reads: “Namely, higher paternal education (OR: 1.21, 95%CI: 1.01-1.44), and higher maternal education (OR: 1.30, 95%CI: 1.05-1.57), together with male gender (OR: 1.50, 95%CI: 1.05-2.15) were positively correlated with appropriate/sufficient PAL at follow-up, altogether evidencing independent influence of predictors on PAL. (Figure 5).” (please see Results section)

Lines 251-253. You mention ‘studies’ but cite only one study

Response: Additional references were added:

  • Moore, L.L.; Lombardi, D.A.; White, M.J.; Campbell, J.L.; Oliveria, S.A.; Ellison, R.C. Influence of parents' physical activity levels on activity levels of young children. The Journal of pediatrics 1991, 118, 215-219.
  • Sallis, J.F.; Patterson, T.L.; Buono, M.J.; Atkins, C.J.; Nader, P.R. Aggregation of physical activity habits in Mexican-American and Anglo families. Journal of behavioral medicine 1988, 11, 31-41.
  • Vilhjalmsson, R.; Thorlindsson, T. The integrative and physiological effects of sport participation: A study of adolescents. Sociological Quarterly 1992, 33, 637-647.
  • Ennis, C.D. Students' Experiences in Sport-Based Physical Education: More Than] Apologies are Necessary. Quest 1996, 48, 453-456.

Text now reads: Previous studies have highlighted three of the most common perspectives for explaining the important determinants of gender-differences in PALs: (i) Socialization, (ii) attitudinal factors, and (iii) organized sports [30,42-45].” (Please see first paragraph of the subsection 4.1; thank you.)

Line 265: I suggest replacing ‘logically’ by ‘consequently’

Response: Thank You, the word logically’ was replaced by ‘consequently’

Lines 307-309: add a reference

Response: Thank you for your suggestion. The reference is added. Please see next comment and response (text is connected)

Sekulic, D.; Sisic, N.; Terzic, A.; Jasarevic, I.; Ostojic, L.; Pojskic, H.; Zenic, N. Sport and scholastic factors in relation to smoking and smoking initiation in older adolescents: a prospective cohort study in Bosnia and Herzegovina. BMJ Open 2017, 7, e014066, doi:10.1136/bmjopen-2016-014066.

Lines 309-313: add a reference

Response: Please see above for reference. Text now reads: “Text now reads: “Specifically, in studied counties, as well as on the whole territory of Bosnia and Herzegovina, the majority of sports are available to all children (i.e., participation is mostly free), the distances between home and sports facilities are relatively short (i.e., children do not depend on parental transport), and most popular sports do not require specific and expensive sports equipment, and are practiced in school facilities and gyms (i.e. team sports like football, handball, basketball) [60].” (Please see 3rd paragraph of the subsection 4.2; Thank you.)

Line 334: add a reference

Response: The reference was added:

  • Andersen, M.R.; Leroux, B.G.; Bricker, J.B.; Rajan, K.B.; Peterson, A.V. Antismoking parenting 541 practices are associated with reduced rates of adolescent smoking. Archives of Pediatrics & Adolescent 542 Medicine 2004, 158, 348-352.; doi: 10.1001/archpedi.158.4.348

Text reads: “Similar findings have been found in studies of some other issues, such as smoking [65].

Lines 340-344: add a reference

Response: Thank you, the reference was added:

  • Zenic, N.; Terzic, A.; Rodek, J.; Spasic, M.; Sekulic, D. Gender-Specific Analyses of the Prevalence and Factors Associated with Substance Use and Misuse among Bosniak Adolescents. Int J Environ Res Public Health 2015, 12, 6626-6640, doi:10.3390/ijerph120606626

Text now reads: The most likely reason for such relative inconsistency in our results (i.e., non-significant influence of parental control and monitoring variables on the PALs of their children) compared to those reported previously, where the authors regularly confirmed significant correlations between similar sets of variables that could be found in the differences between the established “magnitude” of parental control/monitoring [67].” (Please see beginning of the last paragraph of the subsection 4.3; thank you)

Line 368: add the practical applications and the implications for future research

Response: Thank You for your comment. The practical applications and the implications for future research were added and the text now reads: “Findings of this study suggest practical interventions on children with parents of a lower educational level and the need to further examine decline in PAL for girls in earlier age than analyzed here. Also, additional factors related to decreased PAL in boys between 14 and 16 years of age should be evaluated.” (Please see last paragraph of the Discussion section). Also, additional details about practical applicability are provided in the conclusion section. 

Thank you for your suggestions and comments.

Staying at your disposal!

Reviewer 4 Report

The proposed study is relevant; however, it presents important problems in its design and presentation.

In its introduction, information is presented on the practice of Physical Activity and health in young people, PA recommendations and youth habits, the importance of young age in the formation of habits (on this aspect, more development would be lacking).

Studies on the decline in PA practice in adolescents and the identification of the factors that affect PA levels are studied in depth, but what aspects condition the formation of appropriate attitudes and habits for PA practice? If the study is aimed at delving into the analysis of the parent and family variables, it is necessary to delve deeper into what is already known on this issue, for this reason, the introduction should provide more information on this topic.

Here are some of the many possible suggestions:

 Moore, L. L., Lombardi, D. A., White, M. J., Campbell, J. L., Oliveria, S. A., & Ellison, R. C. (1991). Influence of parents' physical activity levels on activity levels of young children. The Journal of pediatrics118(2), 215-219.

Trost, S. G., & Loprinzi, P. D. (2011). Parental influences on physical activity behavior in children and adolescents: a brief review. American Journal of Lifestyle Medicine5(2), 171-181.

Mellin, A. E., Neumark-Sztainer, D., Story, M., Ireland, M., & Resnick, M. D. (2002). Unhealthy behaviors and psychosocial difficulties among overweight adolescents: the potential impact of familial factors. Journal of adolescent health31(2), 145-153.

Simonen, R. L., Perusse, L. O. U. I. S., Rankinen, T., Rice, T., Rao, D. C., & Bouchard, C. (2002). Familial aggregation of physical activity levels in the Quebec Family Study. Medicine and science in sports and exercise34(7), 1137-1142.

The objective must be more precise. A list of the factors that are analyzed in the study is not included. It is necessary to consider include some hypothesis and if it is expected that any of the studied variables will act as an adjustment, mediating or confounding variable.

Method

It should begin by explaining the study design

The instruments used as well as their validity and reliability should be presented and explained.

It should be explained how many institutes participated in the study and how they were selected.

The data analysis should further explain how the multivariate regression model will be applied.

Page 4. lines 139-143 It should be explained which test was used and support it with some bibliographic reference. What do you mean by cluster?

Results

Pag. 5 line 186. Review the interval for boys.

For the application of the multivariate model, it is necessary to include the results of the correlations between the study variables and their interaction. It is mentioned in the data analysis, but not in the results.

Data are missing in the regression studies (beta, R2, t or F value ...)

Discussion

Doubts about all the above do not allow me to analyze this section.

Author Response

The proposed study is relevant; however, it presents important problems in its design and presentation.

In its introduction, information is presented on the practice of Physical Activity and health in young people, PA recommendations and youth habits, the importance of young age in the formation of habits (on this aspect, more development would be lacking).

Studies on the decline in PA practice in adolescents and the identification of the factors that affect PA levels are studied in depth, but what aspects condition the formation of appropriate attitudes and habits for PA practice? If the study is aimed at delving into the analysis of the parent and family variables, it is necessary to delve deeper into what is already known on this issue, for this reason, the introduction should provide more information on this topic.

Thank you for your comments. We tried to follow it strictly and amended the manuscript accordingly. Please see bellow how we tried to improve the manuscript and where to find the specified changes. 

Here are some of the many possible suggestions:

 Moore, L. L., Lombardi, D. A., White, M. J., Campbell, J. L., Oliveria, S. A., & Ellison, R. C. (1991). Influence of parents' physical activity levels on activity levels of young children. The Journal of pediatrics, 118(2), 215-219.

Trost, S. G., & Loprinzi, P. D. (2011). Parental influences on physical activity behavior in children and adolescents: a brief review. American Journal of Lifestyle Medicine, 5(2), 171-181.

Mellin, A. E., Neumark-Sztainer, D., Story, M., Ireland, M., & Resnick, M. D. (2002). Unhealthy behaviors and psychosocial difficulties among overweight adolescents: the potential impact of familial factors. Journal of adolescent health, 31(2), 145-153.

Simonen, R. L., Perusse, L. O. U. I. S., Rankinen, T., Rice, T., Rao, D. C., & Bouchard, C. (2002). Familial aggregation of physical activity levels in the Quebec Family Study. Medicine and science in sports and exercise, 34(7), 1137-1142.

Response: Thank You for your comment. Part of the introduction regarding parental influence on PAL has been rewritten and now reads: “[32]. Of the sociological factors, parental support stands out most consistently as a factor that positively influences PALs due to its positive influence on the involvement of youths in PA [27,34,35]. In this context, parents prove to be more important than other agents of socialization (colleagues, school, etc.) because, in addition to influencing young people as role models, they also serve as a kind of “gate keeper” by enrolling children in sports clubs, driving them to training, and so on [19]. Studies examining parental influence on children’s PAL have identified a number of predictors [28,29,36,37]. Children of active mothers and fathers are proven to be multiple times more active than children of inactive parents [29]. Parental support was consistently positively and significantly associated with child PA in numerous studies examining these factors [28]. Among the overweight children and adolescents, higher levels of family connectedness, parental expectations and moderate levels of parental monitoring were associated with the lower level of PA [36]. Also, familial environmental and genetic factors showed to have significant influence for the familial resemblance in physical activity level [37].

The objective must be more precise. A list of the factors that are analyzed in the study is not included. It is necessary to consider include some hypothesis and if it is expected that any of the studied variables will act as an adjustment, mediating or confounding variable.

Response: Thank You. We amended the objectives of the study and included the specific hypothesis. Text now reads: “For these reasons, the main aim of this study was to prospectively evaluate the changes in PALs among adolescents from Bosnia and Herzegovina over the 2-years period, between 14 and 16 years of age. Further, we evaluated the influence of parental/familial factors on PALs at the beginning of the 1st year of high school (approximately 14 years of age), and at the end of 2nd year of high-school (approximately 16 years of age). Authors hypothesized that decline of PAL will occur during the course of the study. Additionally, we hypothesized that parental/familial factor would influence PALs in both boys and girls.” (Please see last paragraph of the Introduction; Thank you)

Method

It should begin by explaining the study design

Response: Thank You, the text is amended accordingly. In this version of the manuscript we followed your suggestions and also the one suggestion of Reviewer 5 so we adjusted Methods section according to all comments. In this version of the manuscript the subheadings of the Methods are as follows: 2.1. Design and participants; 2.2. Instruments; 2.3. Procedures; 2.4. Data analysis. Thank you!

The instruments used as well as their validity and reliability should be presented and explained.

Response: Thank you for your suggestion. In the revised version of the manuscript, more details on questionnaires are added, and text now reads: “The variables in this study included: (i) Participants’ sociodemographic characteristics, (ii) parental/familial factors, and (iii) PALs. Sociodemographic characteristics and parental/familial factors were evaluated by structured questionnaires which were previously confirmed to be valid and reliable in similar samples of participants, and results are presented in details elsewhere [41-43].” (Please see first paragraph of the Methods section; thank you).

It should be explained how many institutes participated in the study and how they were selected.

Response: Thank you for your suggestion. By all means, in the original version of the manuscript we didn’t pay enough attention on selection of the participants (schools) and overall sampling procedure. Therefore, in this version we tried to explain it more specifically and in more details. Specifically, the text about participants and selection now reads: “The participants in this study were adolescents from Bosnia and Hercegovina, more precisely, from Tuzla county, Herzegovina–Neretva county, and Western Herzegovina county. The sampling was based on a multi-stage cluster sampling method including (i) clustering of all schools from selected counties into two cluster (based on school-size), (ii) random sampling of 50% of high schools from each cluster, and (iii) random sampling of 50% of 1st grades from each of the selected schools. During the baseline testing, a total of 701 participants were examined. Therefore, the inclusion criteria for this study was: (i) regular participation in the high-school education in selected high-schools, and (ii) participation in testing at both testing waves (please see later for details on testing). No specific exclusion criteria were specified.” (please see Begining of the Materials and Methods susbsection). The number of schools included is presented in Figure 1. Thank you!

The data analysis should further explain how the multivariate regression model will be applied.

Response: Thank you for your comment. In the revised version of the manuscript more details are provided and multivariate analyses are explained. Text reads: “To define the influence of studied predictors on PAL at baseline and follow-up the logistic regression was applied. First, each predictor was correlated with dichotomized PAQ-A values (insufficient PAL was coded as “1”, and sufficient PAL was coded as “2”). In order to further evaluate the eventual co-variability of the predictors, and to identify/eliminate any possible causal relationship between predictors, in the last phase of the statistical analyses the multivariate logistic regressions were calculated for dichotomized criteria (PAL at baseline, and PAL at follow-up). For such purpose all predictors found to be significantly correlated to PAL were included in the multivariate logistic regression model. The final model was checked by Hosmer-Lemeshow test of model fit (with significant χ2 indicating inappropriate model fit). Negelkerke R square, p-values, Odds Ratio (OR) and corresponding 95% Confidence Interval (95%CI) were reported as indicators of association between predictors and criteria.” (please see last paragraph of the Statistics subsection; Thank you).

Page 4. lines 139-143 It should be explained which test was used and support it with some bibliographic reference. What do you mean by cluster?

Response: Thank you for your comment. In the revised version of the manuscript we tried to explain this analysis in more details. Also, reference is included as you suggested (Killip, S.; Mahfoud, Z.; Pearce, K. What is an intracluster correlation coefficient? Crucial concepts for primary care researchers. Ann Fam Med 2004, 2, 204-208, doi:10.1370/afm.141.) Text reads: “After conducting both baseline- and follow-up tests, analysis of the attrition bias was calculated No significant differences were evidenced in PAL between the children tested at both waves and the ones who dropped out. However, drop-out rate was higher in males than in females. By calculating intracluster correlation (with schools as clusters) we evidences the relatedness of responses within each cluster (school) [44]. Specifically, intracluster coefficient (IC) for the baseline PAL showed appropriate within-school variance (IC = 0.06).” (please see last paragraph of the Design and participants subsection; Thank you)

Results

Pag. 5 line 186. Review the interval for boys.

Response: Thank you for noticing this mistake. Indeed, we missed to report the follow-up value for boys. It is corrected now, and text reads: “PALs decreased significantly during the course of the study in total sample (from 2.26 ± 1.13 to 2.13 ± 1.06; t-test = 16.89, p < 0.001), and when observed separately for boys (from 2.42 ± 1.19 to 2.28 ± 1.01, t-test = 10.41, p < 0.001), and for girls (from 2.14 ± 1.07 to 2.01 ± 0.99, t-test = 13.42, p < 0.001).” (Please see first paragraph of the Results subsection; thank you)

For the application of the multivariate model, it is necessary to include the results of the correlations between the study variables and their interaction. It is mentioned in the data analysis, but not in the results.

Response: Indeed, in the original version we didn’t specify results on model fit (interaction between predictors); In the revised version we provided details about model-fit. text reads: “Hosmer Lemeshov test evidenced appropriate model fit for multivariate logistic regression models calculated for PAL at baseline (Chi square = 7.37 p = 0.39), and PAL at follow-up (Chi square = 8.01, p = 0.31).” (Please see end of results section)

Data are missing in the regression studies (beta, R2, t or F value ...)

Response: Thank you for your comment. Actually, we evidently missed to sufficiently highlight the numerical values of the logistic regression analyses we calculated. Therefore, in this version of the manuscript we provided additional details including Negelkerke’s R square and p-levels. For example, text for univariate logistics reads: “Figures 3 and 4 present univariate relationships between baseline sociodemographic and parental/familial factor, and dichotomized PAL criterion at baseline (Figure 3) and at follow-up (Figure 4). At baseline, the higher likelihood for appropriate PAL was found in males (Negelkerke R square: 0.05; OR: 1.68, 95%CI: 1.21-2.34; p < 0.001), for those adolescents whose mothers- (Negelkerke R square: 0.04; OR: 1.38, 95%CI: 1.15-1.70; p < 0.001), and whose fathers- were better educated (Negelkerke R square: 0.04; baseline: OR: 1.35, 95%CI: 1.10-1.65, p < 0.01). At follow-up higher likelihood for appropriate PAL was found for boys (OR: 1.54, 95%CI: 1.11-2.03, p < 0.001), adolescents who reported better maternal- (Negelkerke R square: 0.02; OR: 1.35, 95%CI: 1.11-1.69, p < 0.05), and those who reported better paternal-education (Negelkerke R square: 0.02; OR: 1.29, 95%CI: 1.09-1.59, p < 0.05). “ (please see Results).

For multivariate regressions text readsMultivariate logistic regressions was calculated while simultaneously including all variables evidenced as being significant univariate predictors of PAL at baseline and follow-up. In brief, male gender (OR: 1.55, 95%CI: 1.11-1.91, p < 0.001), higher paternal education (OR: 1.35, 95%CI: 1.05-1.67, p < 0.05), and higher maternal education (OR: 1.34, 95%CI: 1.06-1.71, p < 0.05) were all significantly related to PAL-baseline (Negelkerke R square: 0.04). In total 67% of the participants were correctly classified according to specified regression function. Similar multivariate relationship was found when PAL at follow up was observed as criterion. Namely, higher paternal education (OR: 1.21, 95%CI: 1.01-1.44, p < 0.05), and higher maternal education (OR: 1.30, 95%CI: 1.05-1.57, p < 0.01), together with male gender (OR: 1.50, 95%CI: 1.05-2.15, p < 0.01) were positively correlated with appropriate/sufficient PAL at follow-up (Negelkerke R square: 0.05), with 71% participants being correctly classified. Results of the multivariate logistic regressions actually evidence the independent influence of paternal education and gender on PAL (Figure 5).”

Discussion

Doubts about all the above do not allow me to analyze this section.

Thank you for your suggestions and comments.

Staying at your disposal

Reviewer 5 Report

I have enjoyed reading your manuscript, which is interesting, rigorous, and adequate. However, I am going to make some suggestions for improvement that I think can contribute to increasing the quality of the manuscript.

1) Abstract

  • Avoid acronyms that are not necessary in the abstract, for example physical activity level (PAL). I think these acronyms in the body of the text would be more recommended.

2) Introduction and Discussion

Including information related to these jobs can be very helpful. I suggest citing the following articles.

  • Moral-García, J. E., Urchaga-Litago, J. D., Ramos-Morcillo, A. J., & Maneiro, R. (2020). Relationship of Parental Support on Healthy Habits, School Motivations and Academic Performance in Adolescents. International journal of environmental research and public health, 17(3), 882. https://doi.org/10.3390/ijerph17030882

  • Serna, C.; Martínez, I. Parental Involvement as a Protective Factor in School Adjustment among Retained and Promoted Secondary Students. Sustainability 2019, 11(24), 7080; https://doi.org/10.3390/su11247080.
    • https://www.mdpi.com/2071-1050/11/24/7080

  • Garcia, F.; Serra, E.; Garcia, O.F.; Martinez, I.; Cruise, E. A Third Emerging Stage for the Current Digital Society? Optimal Parenting Styles in Spain, the United States, Germany, and Brazil.  J. Environ. Res. Public Health2019, 16, 2333.
    • https://www.mdpi.com/1660-4601/16/13/2333

  • Moral-García, J. E., Agraso-López, A. D., Ramos-Morcillo, A. J., Jiménez, A., & Jiménez-Eguizábal, A. (2020). The influence of physical activity, diet, weight status and substance abuse on students’ self-perceived health. International Journal of Environmental Research and Public Health, 17(4), 1387. https://doi.org/10.3390/ijerph17041387

Classic jobs can also be used:

  • Spera, C. (2005). A review of the relationship among parenting practices, parenting styles, and adolescent school achievement. Educational Psychology Review, 17, 125-146.

  • Steinberg, L., Lamborn, S. D., Dornbusch, S. M., & Darling, N. (1992). Impact of parenting practices on adolescent achievement: Authoritative parenting, school involvement, and encouragement to succeed. Child Development, 63, 1266-1281.

3) Materials and Methods (line 60  to 95). Some suggestions are proposed.

  • Distribute in the following sections: Design and Participants; Instruments; Process; Data analysis.
  • Explain the process of selecting the sample of participants (in participants),
  • Explain the inclusion and exclusion criteria to participate in this study (in procedures).

4) Results

  • They have been addressed correctly.

5) Discussion

  • I consider that the citations used are up to date, but the inclusion of the citations of the aforementioned works is recommended (in point 2. Introduction and Discussion).
  • You can also try to explain the findings by trying to establish more causal relationships between the variables analyzed.

6) Conclusions

  • I think that's fine.

7) Bibliographic references

  • Include the DOI of the manuscripts that have it.

With all the humility these recommendations are collected with the intention that they can be of help to improve this work.

Congratulations on your research.

Author Response

I have enjoyed reading your manuscript, which is interesting, rigorous, and adequate. However, I am going to make some suggestions for improvement that I think can contribute to increasing the quality of the manuscript.

Response: Thank you for your support and suggestions. Please see below for details on amendments.

1) Abstract

Avoid acronyms that are not necessary in the abstract, for example physical activity level (PAL). I think these acronyms in the body of the text would be more recommended.

Response: Thank You. The abstract is amended accordingly, and all abbreviations are ommited

2) Introduction and Discussion

Including information related to these jobs can be very helpful. I suggest citing the following articles.

Moral-García, J. E., Urchaga-Litago, J. D., Ramos-Morcillo, A. J., & Maneiro, R. (2020). Relationship of Parental Support on Healthy Habits, School Motivations and Academic Performance in Adolescents. International journal of environmental research and public health, 17(3), 882. https://doi.org/10.3390/ijerph17030882

Serna, C.; Martínez, I. Parental Involvement as a Protective Factor in School Adjustment among Retained and Promoted Secondary Students. Sustainability 2019, 11(24), 7080; https://doi.org/10.3390/su11247080.

https://www.mdpi.com/2071-1050/11/24/7080

Garcia, F.; Serra, E.; Garcia, O.F.; Martinez, I.; Cruise, E. A Third Emerging Stage for the Current Digital Society? Optimal Parenting Styles in Spain, the United States, Germany, and Brazil.  J. Environ. Res. Public Health2019, 16, 2333.

https://www.mdpi.com/1660-4601/16/13/2333

Moral-García, J. E., Agraso-López, A. D., Ramos-Morcillo, A. J., Jiménez, A., & Jiménez-Eguizábal, A. (2020). The influence of physical activity, diet, weight status and substance abuse on students’ self-perceived health. International Journal of Environmental Research and Public Health, 17(4), 1387. https://doi.org/10.3390/ijerph17041387

Classic jobs can also be used:

Spera, C. (2005). A review of the relationship among parenting practices, parenting styles, and adolescent school achievement. Educational Psychology Review, 17, 125-146.

 Steinberg, L., Lamborn, S. D., Dornbusch, S. M., & Darling, N. (1992). Impact of parenting practices on adolescent achievement: Authoritative parenting, school involvement, and encouragement to succeed. Child Development, 63, 1266-1281.

Response: Thank You for your comment. Most of the suggested references are included in introduction and discussion as it follows:

Introduction:

“Young people with high PALs are less prone to cardiovascular diseases and type 2 diabetes, control their weight better, maintain a healthy musculoskeletal system and respiratory capacities, and also experience mental health benefits (i.e., self-confidence and reduced likelihood of depression) and also have higher level of self-perceived health status [4-11].”

“Studies examining parental influence on children’s PAL have identified a number of predictors [29,30,37-39]. Children of active mothers and fathers are proven to be multiple times more active than children of inactive parents [30]. Parental support was consistently positively and significantly associated with child PA in numerous studies examining these factors [29,39]. Among the overweight children and adolescents, higher levels of family connectedness, parental expectations and moderate levels of parental monitoring were associated with the lower level of PA [37]. Also, familial environmental and genetic factors showed to have significant influence for the familial resemblance in physical activity level [38].”

Discussion:

“Studies find that authoritative parenting styles are often associated with higher levels of student achievement [60].”

“Additionally, in the study examining optimal parent–child relationships parental warmth and strictness was highly associated with child’s well-being [63].”

3) Materials and Methods (line 60 to 95). Some suggestions are proposed.

Distribute in the following sections: Design and Participants; Instruments; Process; Data analysis.

Response: Thank You. The sections are distributed accordingly.

Explain the process of selecting the sample of participants (in participants),

Response: Thank you for your suggestion. The recruitment is explained and text now reads: “The participants in this study were adolescents from Bosnia and Hercegovina, more precisely, from Tuzla county, Herzegovina–Neretva county, and Western Herzegovina county. The sampling was based on a multi-stage cluster sampling method including (i) clustering of all schools from selected counties into two cluster (based on school-size), (ii) random sampling of 50% of high schools from each cluster, and (iii) random sampling of 50% of 1st grades from each of the selected schools.” (please see beginning of the 1st paragraph of the Methods section; Thank you)

Also, the process of recruitment is presented in Figure

Explain the inclusion and exclusion criteria to participate in this study (in procedures).

Response: The inclusion/exclusion criteria are now presented and text reads: Therefore, the inclusion criteria for this study was: (i) regular participation in the high-school education in selected high-schools, and (ii) participation in testing at both testing waves (please see later for details on testing). No specific exclusion criteria were specified.” (Please see 1st paragraph of the Materials and methods subsection; thank you).

4) Results

They have been addressed correctly.

Response: Thank You.

5) Discussion

I consider that the citations used are up to date, but the inclusion of the citations of the aforementioned works is recommended (in point 2. Introduction and Discussion).

Response: Thank you. The recommended references are added and the text is amended accordingly. All details are present in point 2. Introduction and Discussion.

You can also try to explain the findings by trying to establish more causal relationships between the variables analyzed.

Response: Indeed, the causality of the relationship is interesting issue, and we believe that the multivariate design (please see last phase of the statistical analyses) is particularly useful for such purpose. It is now more clearly specified and text reads: “In order to further evaluate the eventual co-variability of the predictors, and to detect any possible causal relationship between predictors, in the last phase of the statistical analyses the multivariate logistic regression was calculated. For such purpose all predictors found to be significantly correlated to PAL were included in the multivariate logistic regression model. The final model was checked by Hosmer-Lemeshow test of model fit. The Odds Ratio (OR) and corresponding 95% Confidence Interval (95%CI) were reported.” (please see last paragraph of the subsection 2.4).

6) Conclusions

I think that's fine.

Response: Thank You.

7) Bibliographic references

Include the DOI of the manuscripts that have it.

Response: Thank You for your comment. DOI is added for each reference.

With all the humility these recommendations are collected with the intention that they can be of help to improve this work.

Congratulations on your research.

Thank you for your suggestions and comments.

Staying at your disposal for any further comments and suggestions

Round 2

Reviewer 4 Report

The work presents considerable improvements. I congratulate the authors for their effort and work

In spite of everything, I suggest paying attention to some aspects that can improve the work presented.

The summary should collect the most important data from the results.

Introduction

The argumentation followed in the introduction continues without developing the fundamental content related to the objective of the study.

What parental and familial factors have been studied? Which of those factors have been most important in predicting physical activity practice?

What instruments have been used?

The objective must refer to socioeconomic, socio-educational and parent-child relationship factors.

Methods

Is your study an experimental longitudinal non-randomized control trial? Please, try to explicitly describe what kind of study design have you used.

The question about the number of institutes from which the study sample came remains unanswered

Results

The effect size given by the r squared value should be analyzed in the discussion

Discussion

The education of the parents is the most important factor for which the authors should make an effort to deepen the possible explanations and the interest of future studies

Author Response

The work presents considerable improvements. I congratulate the authors for their effort and work

In spite of everything, I suggest paying attention to some aspects that can improve the work presented.

RESPONSE: Thank you for recognizing our efforts, and for providing additional comments. We tried to improve the manuscript accordingly. Please see below for responses.

The summary should collect the most important data from the results.

RESPONSE: The abstract is rewritten accordingly and most important results are presented. Please see highlighted text in Abstract.

Introduction

The argumentation followed in the introduction continues without developing the fundamental content related to the objective of the study.

RESPONSE: Indeed, previously we missed to “summarize” all stated in the Introduction and to develop the content of the study. We hope that we succeeded now. Text reads: “Although studies done so far provided evidence on decrease of PAL during period of adolescence on the territory of southeastern Europe  [25,42] , there is limited body of prospective evidence about: (i) changes in PAL which occur in younger adolescents, and (ii) factors which may influence such changes in this territory. Specifically, to the best of our knowledge no study so far prospectively examined the changes in PAL, while examining the socio-economic, socio-educational, and factors of parent-child relationship as covariates of changes in PAL in younger adolescents from the territory of former Yugoslavia.

What parental and familial factors have been studied? Which of those factors have been most important in predicting physical activity practice?

RESPONSE: The details on parental factors which influence PAL are now provided, and text reads: “Studies examining parental influence on children’s PAL have identified a number of predictors, including parental support, family connectedness, parental expectations, and parental monitoring [29,30,37-39]. Children of active mothers and fathers are proven to be multiple times more active than children of inactive parents [30]. Parental support was consistently positively and significantly associated with child PA in numerous studies examining these factors [29,39]. Among the overweight children and adolescents, higher levels of family connectedness, parental expectations and moderate levels of parental monitoring were associated with the lower level of PA [37]. Also, familial environmental and genetic factors showed to have significant influence for the familial resemblance in PAL [38].” (Please see 7th paragraph of the Introduction)

What instruments have been used?

RESPONSE: In this version of the manuscript the instruments used for PAL measurement are explained and text reads: “On the one hand, this can be explained by the various measuring instruments used to estimate PA [28]. In brief, some authors used objective measurement techniques such as heart rate monitors, pedometers and accelerometers [40-43]. Meanwhile, most of the studies which analyzed PALs in children and adolescents used subjective technique methods including self-report questionnaires, interviewer-administered questionnaires, proxy-report questionnaires and diaries [44-50]. Self-report and interviewer-administered questionnaires rely on children's self-reported activity in past period, which can vary from past three days to whole year [44-47]. In proxy reports, parents and teachers provide information regarding children's PAL [48,49]. Finally, diaries were used in only a few studies because it is very demanding for children to take regularly notes about their PAL [50].” (Please see 8th paragraph of the Introduction. Thank you.)

The objective must refer to socioeconomic, socio-educational and parent-child relationship factors.

RESPONSE: Amended accordingly. Objectives now reads: “For these reasons, the main aim of this study was to prospectively evaluate the changes in PALs among adolescents from Bosnia and Herzegovina over the 2-years period, between 14 and 16 years of age. Further, we evaluated the influence of socio-economic, socio-educational, and parent-child relationship factors on PALs at the beginning of the 1st year of high school (approximately 14 years of age), and at the end of 2nd year of high-school (approximately 16 years of age). Authors hypothesized that decline of PAL will occur during the course of the study. Additionally, we hypothesized that studied factors would influence PALs in both boys and girls.” (please see last paragraph of the Introduction).

Methods

Is your study an experimental longitudinal non-randomized control trial? Please, try to explicitly describe what kind of study design have you used.

RESPONSE: Thank you for your suggestion. This was prospective cohort study. It is now clearly stated (please see first sentence of the Materials and Methods section.

The question about the number of institutes from which the study sample came remains unanswered

RESPONSE: Please accept our apology for not specifying it. It is now specified in the Figure 1 (“green” square à 21 schools in total).

Results

The effect size given by the r squared value should be analyzed in the discussion

RESPONSE: The discussion is added and now reads: “Interestingly, even in our study the effect size (based on R square from univariate logistic regressions) for PAL at follow-up was higher for maternal education, than for paternal education (Nagelkerke R square = 0.02 and 0.015, for maternal- and paternal-education, respectively) which indicate stronger influence of maternal education on PAL of the children at the age of 16. Meanwhile, based on same statistical parameter, maternal- and paternal-education are equally important predictors of PAL at baseline (i.e. 14 years of age).” (Please see end of subheading 4.2; Thank you.)

Discussion

The education of the parents is the most important factor for which the authors should make an effort to deepen the possible explanations and the interest of future studies

RESPONSE: Thank you for your suggestion. This idea is promising and certainly deserves attention! We added the text about it in Conclusion. It reads: “Additionally, the established influence of the parental education on PAL of their children should be observed out of the context of relationship. Namely, we can anticipate that specific education of the parents about the benefits and importance of physical activity would be directly translated to PAL of their children. Therefore, we may encourage studies that will investigate the effects of education of the parents (i.e. responsible adults) on their children’s PAL. In doing so special attention should be placed on parents of lower educational status and health-related topics of physical activity.”  Please see highlighted text in Conclusion.

Staying at your disposal

Authors